# Regulation of Insulin Clearance by Non-Esterified Fatty Acids

**DOI:** 10.3390/biomedicines10081899

**Published:** 2022-08-05

**Authors:** Sonia M. Najjar, Raziyeh Abdolahipour, Hilda E. Ghadieh, Marziyeh Salehi Jahromi, John A. Najjar, Basil A. M. Abuamreh, Sobia Zaidi, Sivarajan Kumarasamy, Harrison T. Muturi

**Affiliations:** 1Department of Biomedical Sciences, Heritage College of Osteopathic Medicine, Ohio University, Athens, OH 45701, USA; 2Diabetes Institute, Heritage College of Osteopathic Medicine, Ohio University, Athens, OH 45701, USA; 3Department of Biomedical Sciences, Faculty of Medicine and Medical Sciences, University of Balamand, Balamand P.O. Box 100, Lebanon; 4Department of Internal Medicine, College of Medicine, University of Toledo, Toledo, OH 43606, USA

**Keywords:** insulin resistance, insulin clearance, non-esterified fatty acids, hyperinsulinemia, non-alcoholic fatty liver disease (NAFLD)

## Abstract

Insulin stores lipid in adipocytes and prevents lipolysis and the release of non-esterified fatty acids (NEFA). Excessive release of NEFA during sustained energy supply and increase in abdominal adiposity trigger systemic insulin resistance, including in the liver, a major site of insulin clearance. This causes a reduction in insulin clearance as a compensatory mechanism to insulin resistance in obesity. On the other hand, reduced insulin clearance in the liver can cause chronic hyperinsulinemia, followed by downregulation of insulin receptor and insulin resistance. Delineating the cause–effect relationship between reduced insulin clearance and insulin resistance has been complicated by the fact that insulin action and clearance are mechanistically linked to insulin binding to its receptors. This review discusses how NEFA mobilization contributes to the reciprocal relationship between insulin resistance and reduced hepatic insulin clearance, and how this may be implicated in the pathogenesis of non-alcoholic fatty liver disease.

## 1. Mechanisms of Insulin Clearance

Insulin stimulates fat storage in adipose tissue in addition to increasing glucose uptake and reducing gluconeogenesis in liver. Whereas insulin signaling mediates insulin action, the process is regulated by the circulating level of insulin that is, in turn, determined not only by how much is secreted, but also by how much is cleared (reviewed in [1]).

Insulin is secreted in pulses from pancreatic beta cells [2]. The liver, a major site of insulin clearance from the blood, is the first organ to receive secreted insulin from the portal vein through the fenestrae of its capillaries. Following binding to its receptors on the plasma membrane of hepatocytes, the insulin–insulin receptor complex is targeted to the lysosomal compartments for degradation (reviewed in [1,3,4]). Receptor-mediated insulin uptake and degradation constitutes the basic mechanism of insulin clearance. About 60–70% of insulin undergoes degradation through its first pass through the liver. In this manner, the liver contributes significantly to determining the amount of insulin that reaches peripheral tissues, and as such can regulate their proper response to insulin. Insulin can undergo some degradation in skeletal muscle and adipocytes before returning to the liver for an additional round of degradation. Insulin is terminally degraded in the proximal tubule cells of the kidney [5].

Mechanistically, insulin internalization via its receptor is induced by the Carcinoembryonic Antigen-Related Cell Adhesion Molecule 1 (CEACAM1) (reviewed in [1]). CEACAM1 is a plasma membrane glycoprotein that is ubiquitously expressed with a prominent degree in the liver and to a lower extent in the kidney. In response to pulses of incoming insulin through the portal vein, hepatocytic CEACAM1 undergoes phosphorylation by the insulin receptor tyrosine kinase (Figure 1) on the highly conserved tyrosine 488 residue (Y) to take part of the insulin-receptor endocytosis complex through binding to the SH2 domain of Shc, which in turn binds to tyrosine 960 in the juxtamembrane domain of the β-subunit of the insulin receptor through its phosphotyrosine binding motif (reviewed in [1]). This increases the rate of insulin endocytosis via its high-affinity low dominant receptor A isoform and targeting to the degradation process (Figure 1).

Fatty acid synthase (FASN) binds to CEACAM1 to pull it off and facilitate insulin dissociation from its receptor to allow insulin degradation in the acidic milieu of late endosomes and recycling of the receptor to the plasma membrane (reviewed in [1]). Binding of phosphorylated CEACAM1 to FASN represses its enzymatic activity and limits de novo lipogenesis to prevent fat accumulation in the liver in the face of the high portal insulin levels [6]. In this manner, insulin acutely represses de novo lipogenesis under normal physiologic conditions. We posited that CEACAM1 phosphorylation by pulses of released insulin constitutes a main protective mechanism against the higher levels of insulin in the portal vein, not only by mediating hepatic insulin clearance and maintaining the homeostatic level of insulin, but also by binding to FASN and maintaining low fatty acid synthesis in hepatocytes [6], thus preventing hepatic steatosis. This acute negative effect of insulin on de novo lipogenesis is in contrast to its chronic lipogenic effect whereby hyperinsulinemia, such as in conditions of elevated visceral/abdominal obesity and type 2 diabetes, activates SREBP-1c, a master transcriptional upregulator of the expression of lipogenic genes, including FASN [7]. This in turn leads to increased FASN activity and hepatic steatosis (Figure 2). As abdominal obesity and insulin resistance progress, the pulsatility of insulin release decreases [8,9], leading to defective insulin signaling in hepatocytes, including CEACAM1 phosphorylation and, subsequently, reduction in hepatic insulin clearance to contribute to chronic hyperinsulinemia. This demonstrates that insulin acutely represses lipogenesis and proposes that insulin resistance is inclusive of a defective insulin signaling against de novo lipogenesis as well as gluconeogenesis. This challenges the well-accepted notion of the selectivity of hepatic insulin resistance [10] that comprises a blunted acute effect of insulin on gluconeogenesis with a more positive effect of chronic hyperinsulinemia on lipogenesis.

Additionally, CEACAM1 directly regulates insulin signaling. The insulin–insulin receptor complex formation and internalization are rapidly followed by binding of CEACAM1 to SHP2 phosphatase to sequester it and sustain persistent phosphorylation of insulin receptor substrate-1 and -2 (IRS-1 and IRS-2), key insulin signaling molecules [1,11]. IRS-1/2 signals through the PI3 kinase to maintain insulin action as vesicular insulin degradation occurs. As previously discussed [1], the coordinated binding of FASN to the same phosphorylated tyrosine residue (Y488) at which SHP2 binds to CEACAM1 positions CEACAM1 phosphorylation in response to an acute rise in insulin as a unifying mechanism regulating insulin sensitivity and clearance in hepatocytes. This is consistent with the dominant expression of insulin receptor, CEACAM1, and FASN in hepatocytes that are rapidly exposed to acute rises of insulin in the portal vein through the fenestrae of the liver capillaries, as opposed to other peripheral tissues in which the insulin passage is tightly regulated by the endothelial cells of their vasculature [12].

These early observations in cell systems are bolstered by the manifestation of reduced insulin clearance followed by chronic hyperinsulinemia and its downregulatory effect on insulin receptors to cause hepatic insulin resistance in mice with null mutation or liver-specific *Ceacam1* gene deletion, and in mice with liver-specific inactivation of CEACAM1 (reviewed in [1]). These mice also exhibit chronic hyperinsulinemia-driven hepatic steatosis and other features of non-alcoholic fatty liver disease (NAFLD) (reviewed in [1]).

Although Insulin Degrading Enzyme (IDE) plays a significant role in renal insulin clearance, its role in hepatic insulin clearance remains controversial [13,14]. Studies in mice with liver-specific deletion of *Ide* gene showed controversial results. When mice were propagated on C57BL/6J, they displayed primarily insulin resistance without significant effect on insulin clearance [15]. However, when mice were propagated on the C57BL/6N genetic background [16], deleting *Ide* from the liver led to a primary defect in postprandial insulin clearance. In support of the latter, IDE polymorphisms have been identified in a cohort of Portuguese normoglycemic men in strong association with abnormal postprandial insulin clearance [16]. At the cellular level, IDE may play a role in intracellular trafficking of insulin and, likely, in insulin receptor recycling to the plasma membrane of hepatocytes, as we have recently proposed [1].

## 2. Reduced Insulin Clearance in Metabolic Disease

Hepatic insulin clearance serves as a major determinant of insulin sensitivity in the normal dog [17]. Genome-wide linkage analysis has shown that compromised insulin clearance is a highly heritable trait [18]. Reduction in insulin clearance has emerged as a major risk factor in metabolic syndrome, including NAFLD, in Hispanics (particularly of Mexican descent [19]), African-Americans [20], African-Caribbeans [21], and Native Americans [22]. Reduced insulin clearance has also been increasingly identified in young/adolescent subjects with obesity and NAFLD [23,24]. Hence, it has become imperative to delineate its mechanistic role in the pathogenesis of insulin resistance.

Most studies in animals [25,26] and humans [27] have shown that hepatic insulin clearance is defective in obesity and insulin resistance states [26] (Figure 2). Several animal models have shown that reduced hepatic insulin clearance occurs as an early homeostatic mechanism causing hyperinsulinemia during obesity [28,29,30,31,32,33,34]. In this regard, some studies have attributed the reduction in insulin clearance rate in obese subjects to insulin resistance rather than to excessive adiposity [35]. Nonetheless, reduced insulin clearance predictably limits the dependence on the compensatory increase in insulin secretion in order to spare the functionality of β-cells [36]. On the other hand, insulin clearance, especially post-prandial, has been shown to be higher in subjects with type 2 diabetes than those without the disease [37]. This appears to be related to defective insulin secretion and its resultant postprandial insulin delivery rate to its clearance sites.

Reduced insulin clearance has also been reported to occur as fat deposition increases in the liver, supporting its significant role in chronic hyperinsulinemia in non-alcoholic fatty liver disease (NAFLD), a metabolic disorder that is commonly associated with insulin resistance [38,39]. For instance, hyperinsulinemia appears to be mainly driven by reduced insulin clearance as NAFLD progresses [40]. Furthermore, studies in Japanese patients with type 2 diabetes linked the defect in hepatic insulin clearance to increased degree of hepatic steatosis rather than to visceral adiposity per se [41]. In a cohort of obese South Korean patients, CEACAM1, a major player in promoting insulin clearance, was reported to be progressively reduced with increased hepatic steatosis independently of type 2 diabetes [42]. CEACAM1 protein content is also reduced in the liver and in primary hepatocytes of obese relative to age-matched, middle-aged, lean, non-diabetic men [43].

In patients with type 2 diabetes, the incidence of hepatic steatosis is higher and of insulin clearance is lower than nondiabetic subjects [44]. Lowering of hepatic triglyceride content by administering PPARγ agonists for 16 weeks restores insulin clearance without causing significant weight loss [45]. Given that PPARγ upregulates Ceacam1 transcription [46], it is likely that the beneficial effect of PPARγ activation is mediated by inducing hepatic CEACAM1 expression. This warrants further investigation.

Insulin-resistant patients with type 2 diabetes commonly develop upper central body obesity (i.e., abdominal adiposity) [47,48,49,50,51]. Systemic insulin resistance develops when adipocytes in white adipose tissue (WAT) become insulin-resistant, a process manifested by the release of non-esterified fatty acids (NEFA) [50,52] and adipokines together with reduced production of adiponectin [53,54,55]. Increased visceral obesity also leads to increase leptin production and secretion. If uncontrolled, these factors combined lead to systemic insulin resistance [56,57,58]. The mechanistic basis for the causative role of adipokines in blunting insulin signaling and ensuing insulin resistance has been extensively reviewed [55]. How reduced adiponectin production mediates insulin resistance has also been widely investigated [59]. We herein focus on the role of elevated plasma NEFA in insulin clearance, mainly in the liver, and how this is linked mechanistically to insulin resistance.

## 3. Elevated Plasma NEFA Play a Primary Role in Hepatic Insulin Resistance Followed by Reduced Insulin Clearance

In the Western world, high energy supply, sleep deprivation [60], sedentary lifestyle, and other environmental stressors [61] have been blamed for the rapid increase in upper central body obesity (abdominal obesity), particularly among individuals who are genetically predisposed to type 2 diabetes and NAFLD. The increase in NAFLD in neonates in both humans and animals has been largely attributed to their exposure to high levels of NEFA in utero [62,63].

Chylomicron-triglycerides derived from excessive fat intake that escape β-oxidation are stored in adipose tissue until they are mobilized as NEFA during lipolysis under the effect of catecholamines [64]. The rise in lipolysis-derived NEFA contributes to the deleterious effect of abdominal obesity on insulin sensitivity in skeletal muscle and liver, including elevation in de novo lipogenesis [47,65]. Consistently, inhibiting lipolysis with acipimox restores the suppression of gluconeogenesis in response to insulin without affecting adipokines or adiponectin in subjects at risk of type 2 diabetes [66]. Furthermore, Rosso et al. [67] have shown that lipolysis-derived NEFA play a key role in hepatic inflammation and fibrosis in patients with NAFLD independently of obesity and type 2 diabetes. Moreover, inducing plasma NEFA by intralipid–heparin infusion suppresses insulin clearance in adult women [68] and in normal dogs [69]. Similarly, portal vein infusion of NEFA lowers hepatic insulin clearance and interferes with insulin action in aged rats [70]. On the other hand, caloric restriction improves hepatic insulin clearance and insulin sensitivity in parallel to causing a loss of abdominal fat in adult women [71].

Mechanistically, NEFA can activate PKC (PKCδ [72,73], PKCε [74], and PKCλ [75]) and JNK pathways. Intralipid plus heparin co-infusion in rats has demonstrated that elevated NEFA activation of PKCδ impairs insulin signaling via activating NADPH oxidase to promote oxidative stress and, subsequently, IKK/JNK pathways [72]. JNK activation produces TNFα that could, in turn, blunt insulin signaling by causing serine phosphorylation of IRS-1 [76].

The adverse effect of NEFA on insulin signaling in liver causes reduction in CEACAM1 expression level and phosphorylation and, subsequently, limits hepatic insulin clearance [72] (Figure 3). NEFA can also interfere with insulin degradation by inhibiting insulin binding to its receptor and its cellular uptake/degradation, as shown in isolated rat hepatocytes [77,78]. This provides a mechanistic underpinning for reduced hepatic insulin clearance and insulin signaling by the rise in plasma NEFA under conditions of excessive energy supply and visceral obesity.

Experimental rodents have helped dissect out the mechanisms involved in the progression of insulin resistance in response to high-fat diet. Feeding C57BL/6J mice with a high-fat diet for a prolonged period of time (>3 months) causes systemic insulin resistance driven by a low-grade sub-acute pro-inflammatory state that develops in parallel to the activation of WAT-associated macrophages [53,54,55,79]. However, we [25] and others [80,81] have shown that insulin resistance in liver begins shortly after the initiation of high fat intake in a linear relationship with increased lipolysis and NEFA mobilization out of adipocytes, independently of inflammation. This early lipolysis, resulting from dysregulated hypothalamic control [82], causes hepatic insulin resistance (portal hypothesis) [83,84,85] and blunts insulin signaling by activating PKC pathways, among other mechanisms (Figure 3). As the nutritional burden persists, hepatic steatosis and lipotoxicity develop in addition to insulin resistance in WAT, followed by systemic insulin resistance in parallel with the progression of the pro-inflammatory state and sustained lipolysis [81] (Figure 3). This is consistent with the notion that insulin resistance stems from increased release of NEFA and adipokines from WAT [84,85].

As commonly observed in obese humans [86] and rodents [87,88] with low-grade insulin resistance, adipocytes expand to accommodate fat storage. In such cases of uncomplicated abdominal obesity, lipolysis-derived NEFA in the portal vein [50,52] are mainly removed by fatty acid β-oxidation in the liver [89,90]. Consistently, we have shown that high fat intake causes a progressive reduction in hepatic CEACAM1 levels [25] (Figure 3). This would relieve FASN activity from the suppression by phosphorylated CEACAM1 [6] to produce palmitic acid, which undergoes elongation and unsaturation to eventually yield long chain fatty acyl-CoA, which serve as endogenous ligands of PPARα [91]. With resultant reduction in malonyl-CoA levels and relieving carnitine palmitoyltransferase1 from its inhibitory effect, this bestows a positive feedback mechanism on fatty acid β-oxidation in hepatocytes.

Given that insulin induces the transcriptional activity of Ceacam1 promoter [46], it is likely that NEFA activation of PKC causes the early reduction in Ceacam1 expression via its adverse effect on insulin signaling [72] shortly after high fat intake begins [25] (Figure 3). As stated above, this would initially provide a positive feedback mechanism on fatty acid β-oxidation in liver. Combined with sustained ligand-activated PPARα, which in turn binds to the Ceacam1 promoter to repress its transcription [92], this results in a progressive drop of Ceacam1 expression reaching >50% loss after 3–4 weeks of high fat intake [25]. As previously shown [25], the loss of CEACAM1 by >50% lowers hepatic insulin clearance to contribute to chronic hyperinsulinemia and its downregulatory effect on insulin receptor levels to sustain hepatic insulin resistance. Chronic hyperinsulinemia also causes the transcriptional activation of SREBP-1c to increase lipogenic gene expression [7] and favor de novo lipogenesis over fatty acid β-oxidation. Increased de novo lipogenesis does not only result in hepatic steatosis, but also in VLDL-triglyceride (VLDL-TG) repartitioning to WAT to induce visceral obesity and mediate a vicious cycle of NEFA mobilization along the adipocytes–hepatocytes axis. Eventually, chronic hyperinsulinemia causes insulin resistance in WAT by reducing Glut4-mediated glucose transport [93,94]. The concomitant increase in the pro-inflammatory state in this tissue leads to the release of adipokines and ensuing systemic insulin resistance.

Thus, mobilized NEFA from WAT blunt insulin signaling and activate PPARα to lower Ceacam1 transcription [92]. In this manner, the moderate loss of hepatic Ceacam1 expression (by <50%) by NEFA release before overt insulin resistance develops in adipocytes (such as the case in uncomplicated moderate obesity—[89]) provides a positive feedback mechanism on fatty acid β-oxidation to limit hepatic steatosis in the early phases of increased energy intake and before chronic hyperinsulinemia develops [95]. Whereas the loss of Ceacam1 by <50% promotes fatty acid β-oxidation, its progressive loss by >50% (after about 3–4 weeks) represses hepatic insulin clearance to cause chronic hyperinsulinemia and ensuing hepatic steatosis and hepatic insulin resistance (Figure 3). Prolonged fat intake (for 3–4 months) results in insulin resistance in adipocytes with ensuing systemic insulin resistance.

Nicotinic acid causes expansion of adipocytes to facilitate fat storage in WAT. Despite fat accumulation, inflammation, and blunted insulin signaling in WAT of mice fed a high-fat diet, nicotinic acid treatment protects hepatic CEACAM1 expression in parallel with insulin sensitivity [95]. This supports the findings that lipolysis-derived NEFA repress hepatic CEACAM1 expression, consistent with findings in rats exposed to acute intralipid infusion [72]. Accordingly, hepatic insulin clearance and insulin sensitivity were restored in diet-induced mouse models of obesity undergoing Vertical Sleeve Gastrectomy in parallel to inducing the expression of CEACAM1 and IDE, and independently of weight loss [96].

That reduced hepatic CEACAM1 expression plays a key role in diet-induced hepatic insulin resistance and steatosis in addition to disturbed insulin clearance is supported by the reversal of these metabolic abnormalities upon adenoviral-mediated delivery of the wild-type, but not the phosphorylation-defective, mutants of CEACAM1 to the liver of mice at 14 days of a high-fat diet [95]. Furthermore, forced liver-specific overexpression of CEACAM1 driven by fatty acid-activated human apolipoprotein A1 promoter protects systemic insulin sensitivity in parallel to hepatic insulin clearance [25] as well as adipocyte plasticity [97], to accommodate fat storage in response to prolonged (4 months) high-fat feeding.

Similarly, high-fat feeding causes reduction in hepatic insulin clearance in parallel to reduction in the protein levels of CEACAM1 and IDE [98]. Treatment with rimonabant, a cannabinoid receptor 1 antagonist, reversed hepatic insulin resistance, likely via upregulating adiponectin receptors in the liver. This led to normalization of hepatic insulin clearance via activating CEACAM1 and IDE pathways. It also induced fatty acid β-oxidation while reducing hepatic steatosis.

Of note, not all studies show reduction in insulin clearance in the presence of elevated plasma NEFA levels. For instance, high carbohydrate intake has been shown to result in a marked reduction in plasma NEFA while reducing hepatic insulin clearance and action [99,100]. Moreover, triglyceride emulsion alone does not affect clamp insulin clearance across the hepatic artery and vein [101], whereas it reduces it when glucose is co-infused [68].

## 4. Reduction in Hepatic Insulin Clearance Plays a Primary Role in Insulin Resistance Independently of NEFA Release

The evidence that reduced hepatic insulin clearance constitutes an immediate response to increased dietary energy is mounting in humans and rodents. Several studies have shown that reduced insulin clearance is an early adaptation to increased high energy dietary intake, and that it serves to trigger hyperinsulinemia before insulin resistance, obesity, and increased insulin secretion develop [17,26,100,102]. Feeding healthy, lean European subjects a carbohydrate-rich diet caused a decline in insulin clearance within 3 days [99]. When a group of age-matched healthy, young, lean men were fed a fat- and a carbohydrate-rich Western diet, South Asians who manifested a lower baseline insulin clearance, but not their Caucasian counterparts, developed insulin resistance within 5 days of dietary intake [27]. In healthy, non-obese, Japanese men, low insulin clearance rate is thought to be an early adaptation to maintain normal metabolism and counter adiposity-driven modest impairment of insulin resistance in skeletal muscle [103]. Furthermore, using a systemic-portal insulin infusion method, Bergman et al. [104,105] have hypothesized that reduced hepatic insulin clearance can lead not only to hyperinsulinemia and insulin resistance, but also to increased risk of type 2 diabetes.

Our focus on CEACAM1 phosphorylation by the insulin receptor and its role in insulin clearance has provided a mechanistic underpinning for how reduced insulin clearance can cause insulin resistance and is not just a consequence thereof (Figure 4). Consistent with CEACAM1′s positive role on insulin uptake via its receptor and targeting to its degradation pathways (reviewed in [1]), mice with liver-specific *Ceacam1* deletion display primary reduction in insulin clearance and hyperinsulinemia at about 2 months of age, followed by hepatic insulin resistance (partly caused by downregulation of insulin receptor levels in hepatocytes) and steatohepatitis at 6–7 months of age [106]. Chronic hyperinsulinemia induces de novo lipogenesis (driven by elevated transcriptional activation of SREBP-1c [7]), followed by redistribution of the resultant VLDL-triglyceride (VLDL-TG) to WAT for storage. This is also followed by blunting of insulin signaling in the hypothalamus, a process that activates FASN to cause hyperphagia. Hyperphagia contributes to visceral obesity (together with increased VLDL-TG repartitioning to WAT) and to systemic insulin resistance, as shown by pair-feeding experiments [106]. Visceral obesity eventually leads to insulin resistance in WAT at about 9 months of age, with release of adipokines and lipolysis-derived NEFA, both of which ultimately cause systemic insulin resistance [106]. According to this paradigm, reduced hepatic insulin clearance preceded hepatic insulin resistance and steatosis that led to visceral obesity and insulin resistance in WAT, and eventually adipokine and NEFA release, to cause systemic insulin resistance (Figure 4).

Further buttressing the primary role of hepatic insulin clearance in insulin resistance, global *Ceacam1* null mice manifest reduction in insulin clearance at 2 months of age, followed by systemic insulin resistance at 6–7 months of age and hepatic steatosis. Nicotinic acid treatment stops lipolysis but fails to curb insulin resistance [95]. This demonstrates that reduced hepatic insulin clearance causes insulin resistance together with hepatic steatosis, independently of visceral obesity and lipolysis.

Reversal of the metabolic phenotype upon liver-specific reconstitution of CEACAM1 further demonstrates the primary role of hepatic insulin clearance in the pathogenesis of insulin resistance [107]. Interestingly, high-fat feeding causes a progressive NASH phenotype in global *Ceacam1* null mice, including a higher degree of macro-steatosis in parenchyma, and more robust hepatic fibrosis and apoptosis [108]. This advanced phenotype is prevented by rescuing CEACAM1 in the liver [108], providing further in vivo demonstration of the critical role that CEACAM1-dependent insulin clearance pathways play in the pathogenesis of hepatic insulin resistance and NAFLD/NASH.

## 5. Concluding Remarks

Reduced insulin clearance has increasingly been recognized as an integral part of insulin resistance, irrespective of whether it plays a causal or an adaptive role in insulin resistance. Dissecting out the relationship between insulin action and insulin clearance has been challenged by the dependence of both processes on insulin binding to its receptor and insulin signaling. Post-receptor mechanisms such as phosphorylation of CEACAM1 also regulate insulin signaling (by sequestering SHP2 phosphatase and activating the IRS/PI3Kinase/AKT pathway) (reviewed in [1]), in addition to upregulating insulin clearance. Given the rise in abdominal obesity in parallel to insulin resistance and NAFLD, attention has been given to the significant role that a lipolysis-driven rise in plasma NEFA plays in linking hepatic insulin resistance to reduced insulin clearance. We herein reviewed how the rise in plasma NEFA is implicated in the regulation of insulin action along the adipocytes–hepatocytes axis in response to dietary fat intake, and reciprocally along the hepatocytes–adipocytes axis as driven by genetic predisposition to lower insulin clearance (such as subjects with lower baseline hepatic insulin clearance). Whereas the former paradigm has been widely investigated, more studies are needed to delineate the underlying mechanisms of the second paradigm and to investigate whether decreased hepatic CEACAM1 levels are implicated in genetic predisposition to the disease.

## Figures and Tables

**Figure 1 biomedicines-10-01899-f001:**
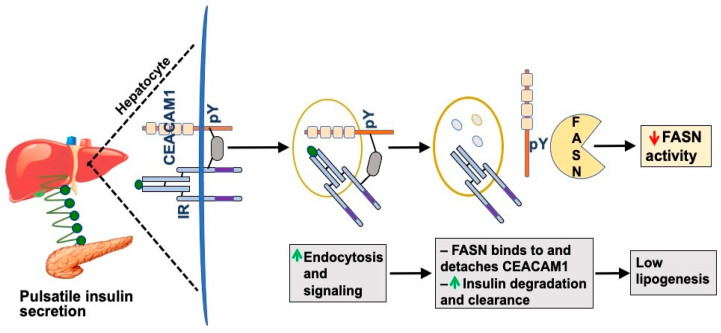
CEACAM1 phosphorylation induces hepatic insulin clearance. In response to pulses of secreted insulin from pancreatic β-cells (green circles), the insulin receptor (IR) tyrosine kinase on the surface membrane of hepatocytes is activated. This phosphorylates CEACAM1 on the species conserved tyrosine (pY) 488 residue, an event that mediates the formation of a stable insulin–IR–CEACAM1 complex to increase the rate of insulin endocytosis and targeting to its degradation process (green upward arrow). Fatty acid synthase (FASN) binds to pY488 of CEACAM1 to cause its detachment and facilitate the separation of insulin from its receptor to undergo degradation in the acidic milieu of late endosomes (grey circles). FASN binding to phosphorylated CEACAM1 also causes repression of FASN activity (red downward arrow), thus maintaining low lipogenesis in the liver despite the higher level of insulin in the portal vein than systemic circulation. Not shown in this schematic diagram, IR recycles back to the surface membrane.

**Figure 2 biomedicines-10-01899-f002:**
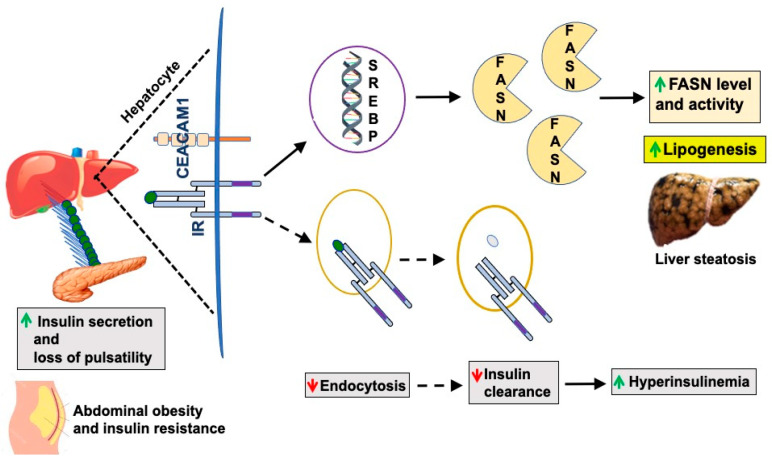
Elevated de novo lipogenesis in obesity. Under conditions of abdominal obesity and insulin resistance, compensatory insulin secretion increases (green upward arrow) and pulsatility of insulin release is lost. This compromises insulin signaling, endocytosis (dotted lines), and degradation (red downward arrow). The resultant reduction in insulin clearance contributes to chronic hyperinsulinemia (green upward arrow), which, in turn, activates SREBP-1c to induce FASN expression and, subsequently, lipogenesis and hepatic steatosis.

**Figure 3 biomedicines-10-01899-f003:**
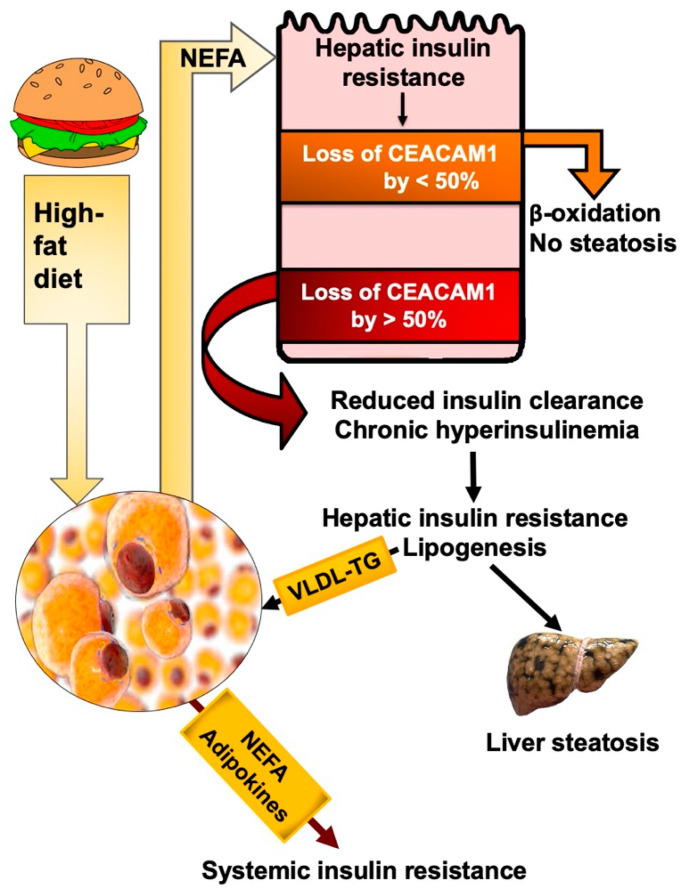
NEFA play a primary role in reducing insulin clearance. High-fat feeding leads to visceral obesity with increased NEFA release into the portal vein even before insulin resistance develops in the adipose tissue. This activates PKC-JNK pathways to cause hepatic insulin resistance and ensuing decrease in Ceacam1 expression. Reduction in Ceacam1 by <50% provides a positive feedback mechanism on fatty acid β-oxidation. When the loss of Ceacam1 reaches >50%, insulin clearance is reduced and chronic hyperinsulinemia ensues, followed by increased lipogenesis and VLDL-triglyceride (VLDL-TG) redistribution to white adipose tissue to trigger sustained NEFA and adipokine release and, subsequently, systemic insulin resistance.

**Figure 4 biomedicines-10-01899-f004:**
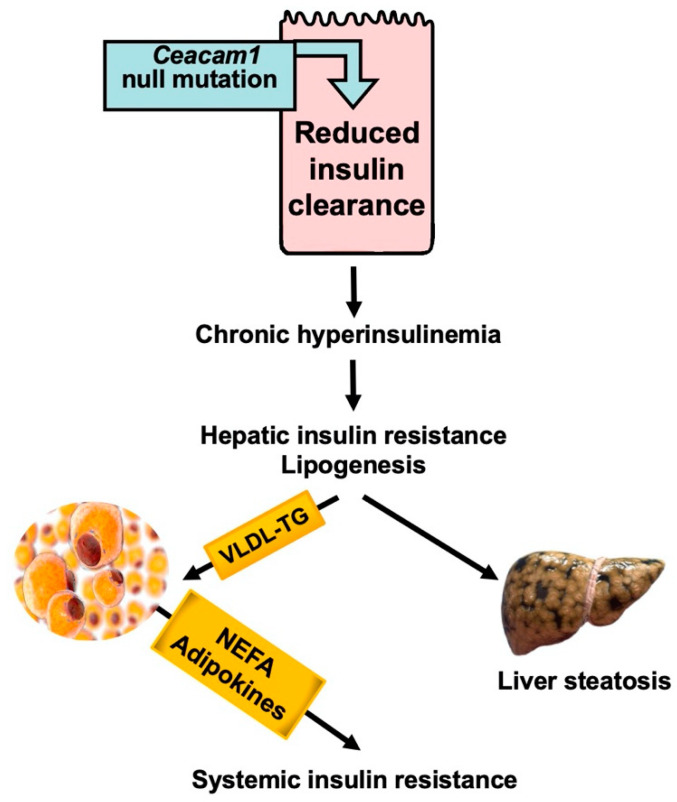
Reduced hepatic insulin clearance causes hepatic insulin resistance independently of NEFA release. Liver-specific Ceacam1 null mutation causes reduction in insulin clearance which, in turn, leads to chronic hyperinsulinemia. This drives hepatic insulin resistance and lipogenesis, followed by redistribution of VLDL-TG to WAT and lipolysis.

## Data Availability

Not applicable.

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
