# Peer review of "Regulation of Insulin Clearance by Non-Esterified Fatty Acids"

_biomedicines, 2022, doi:10.3390/biomedicines10081899_

Round 1

Reviewer 1 Report

It is a well-conducted study and well-written manuscript thus I recommend this manuscript as a suitable review for publications in the “Biomedicines”, with minor comments:

- The abstract is incomplete. The authors should address in the Abstract a brief description of the current knowledge regarding the relationship between insulin resistance - insulin clearance- non-esterified fatty acids.

- Please add a reference in line 39.

- Kindly add research limitations, if any, in the “Conclusion’’ section.

Author Response

Reviewer 1

- The abstract is incomplete. The authors should address in the Abstract a brief description of the current knowledge regarding the relationship between insulin resistance - insulin clearance- non-esterified fatty acids.

Najjar: This was done. We added statements in lines 21-23

- Please add a reference in line 39.

Najjar: We herein added 3 new references (1, 3, 4). Please note that the line is now 43.

- Kindly add research limitations, if any, in the “Conclusion’’ section.

Najjar: These discussions are made throughout the manuscript. Our responses to Reviewer 2 added to these discussions.

Reviewer 2 Report

General comments:

This review focuses on the role of NEFA in regulating insulin clearance - insulin resistance, focusing on the relevance of CEACAM1 in this relationship.

Strengths:

This is a timely and important review given that Insulin clearance and insulin resistance are key mechanisms in diabetes, obesity, and fatty liver. In particular, in diabetes, that classically has been attributed to alterations in glucose metabolism, and for which knowledge of other implicated mechanisms is relevant, since insulin has metabolic actions in other pathways besides glucose metabolism, and metabolism in the body occurs in an integrated way.

The article is well written, and the authors address a complex topic clearly.

1. Readers of the article would benefit from it containing a figure illustrating all the metabolic pathways addressed in an integrated way. Since these pathways are present under normal conditions, a figure comparing health and disease states would help the reader better understand the paper.

2. The authors state, "In this manner, acutely insulin suppresses de novo lipogenesis under normal physiologic conditions. “ (ln59). In addition, insulin acts through other pathways, namely the IRS/PI3K/AKT pathway, addressed by the authors in the following paragraph. There is also the concept that hepatic insulin resistance is selective, with altered suppression of neoglucogenesis but not lipogenesis. Thus, given that the effect on lipogenesis exists in both of these pathways, the question arises whether this is a negative feedback mechanism and if this argues against selective insulin resistance in the liver. The authors could expand on this. Additionally, commenting on the impact of NEFA through CEACAM integrating both pathways (inhibition of FASN and IRS1-2/PI3K/AKT). 

3. In line 116, the authors state, "In patients with type 2 diabetes, the incidence of hepatic steatosis and insulin clearance is lower than in non-diabetic individuals."

There is conflicting evidence as to whether insulin clearance is increased or decreased in individuals with type 2 diabetes, which is further complicated by biases as diabetes medication. It may also depend on the subtype of type 2 diabetes since it is increasingly recognized that type 2 diabetes is very heterogeneous. In addition, the lower insulin clearance in type 2 diabetes may be due to the increased hepatic steatosis, considering that in the cited work, the authors did not demonstrate its independence in people with type 2 diabetes. The authors may clarify this point. 

Author Response

Reviewer 2

- Readers of the article would benefit from it containing a figure illustrating all the metabolic pathways addressed in an integrated way. Since these pathways are present under normal conditions, a figure comparing health and disease states would help the reader better understand the paper.

Najjar: We have added Figure 1 (for normal) and Figure 2 for the disease state.

-The authors state, "In this manner, acutely insulin suppresses de novo lipogenesis under normal physiologic conditions. “ (ln59). In addition, insulin acts through other pathways, namely the IRS/PI3K/AKT pathway, addressed by the authors in the following paragraph. There is also the concept that hepatic insulin resistance is selective, with altered suppression of neoglucogenesis but not lipogenesis. Thus, given that the effect on lipogenesis exists in both of these pathways, the question arises whether this is a negative feedback mechanism and if this argues against selective insulin resistance in the liver. The authors could expand on this. Additionally, commenting on the impact of NEFA through CEACAM integrating both pathways (inhibition of FASN and IRS1-2/PI3K/AKT). 

            Najjar: We have discussed now in lines 84-97 and 110-117 two paragraphs to expand on the innovative hypothesis that acutely insulin represses FASN activity.

-In line 116, the authors state, "In patients with type 2 diabetes, the incidence of hepatic steatosis and insulin clearance is lower than in non-diabetic individuals."

            Najjar: we appreciate why the reviewer made this comment. But, actually we have stated the following “In patients with type 2 diabetes, the incidence of hepatic steatosis is higher and of insulin clearance is lower than nondiabetic subjects [53].”. This statement is now in line 167-168

-There is conflicting evidence as to whether insulin clearance is increased or decreased in individuals with type 2 diabetes, which is further complicated by biases as diabetes medication. It may also depend on the subtype of type 2 diabetes since it is increasingly recognized that type 2 diabetes is very heterogeneous. In addition, the lower insulin clearance in type 2 diabetes may be due to the increased hepatic steatosis, considering that in the cited work, the authors did not demonstrate its independence in people with type 2 diabetes. The authors may clarify this point.

            Najjar: In line, 152-155, we have now added a statement that reads “On the other hand, insulin clearance, especially post-prandial, has been shown to be higher in subjects with type 2 diabetes than those without the disease [46]. This appears to be related to defective insulin secretion and its resultant postprandial insulin delivery rate to its clearance sites.”
